# Chlorogenic Acid and VX765 Alleviate Deoxynivalenol-Induced Enterohepatic Injury and Lipid Metabolism Disorders by Improving Intestinal Microecology

**DOI:** 10.3390/toxins17090467

**Published:** 2025-09-18

**Authors:** Tao Wen, Sirui Li, Huijun Guo, Jinbo Wang, Xinru Zhang, Chunyang Wang

**Affiliations:** Shandong Provincial Key Laboratory of Zoonoses, College of Animal Veterinary Medicine, Shandong Agricultural University, Tai’an 271018, China; 2023120664@sdau.edu.cn (T.W.); 2023120624@sdau.edu.cn (S.L.); hjguo@sdau.edu.cn (H.G.); w15145382416@163.com (J.W.); 2023110395@sdau.edu.cn (X.Z.)

**Keywords:** deoxynivalenol, chlorogenic acid, VX765, microbiota-gut-liver axis, lipid metabolism

## Abstract

Widespread contamination of deoxynivalenol (DON) in cereals and feed threatens global food safety. This study investigated the effects of Chlorogenic acid (CGA) and VX765 on DON-induced enterohepatic injury. A total of 48 female mice were divided into four groups: control (normal saline), DON (1 mg/kg.bw), CGA (100 mg/kg.bw CGA + 1 mg/kg.bw DON), and VX765 (100 mg/kg.bw VX765 + 1 mg/kg.bw DON). After 28-day gavage period, the results showed that CGA and VX765 reduced DON-induced intestinal barrier damage. Metabolomics data revealed that CGA and VX765 restored cecal microbiota structure and alleviated DON-induced hepatic injury and lipid metabolic disorders by reshaping intestinal microbiota. Retrograde endocannabinoid signaling was identified as a critical pathway for cecal microbial metabolism and hepatic lipid regulation mediated by CGA and VX765. Additionally, CGA and VX765 reversed the upregulation of *IMPA*, *CDS2*, *DGKA*, *NDUFS8*, and *MAPK1* mRNA and protein expression levels induced by DON via the microbiota-gut-liver axis.

## 1. Introduction

Deoxynivalenol (DON), a member of the type B trichothecenes, is commonly found in cereals, feed materials and natural environment [1]. DON exhibits remarkable chemical stability, making it challenging to eliminate from contaminated agricultural products through conventional processing methods. DON can readily transfer to meat, eggs, and milk through the food chain, threatening human food safety and animal feed [2,3]. Most monogastric animals are extremely sensitive to DON during both chronic and subchronic exposure [4,5]. After ingestion, DON is absorbed by the intestinal tract, transported to the liver via blood circulation, and eventually excreted into the kidneys through urine following a series of metabolic processes [6]. This can disrupt the normal function of multiple organs and tissues, mainly including the gastrointestinal tract, liver, kidney and brain [7,8,9].

With the extensive and in-depth investigation of the animal microbiome, a growing body of research has underscored the existence of a complex, bidirectional relationship between gut microbes and peripheral organs, often termed an “axis” [10,11]. As integral components of the digestive system, the liver and intestine engage in extensive and intricate bidirectional communication through the gut-liver axis [12]. This axis plays a critical role not only in fat deposition, energy balance, nutrient metabolism, and immune function but also in mediating host-environment interactions [13,14]. In these processes, gut microbes and their metabolites act as key mediators. Researchers have proved that intestinal icrobial dysbiosis and compromised intestinal barrier can contribute to a spectrum of liver diseases and metabolic disorders via gut–liver axis [15,16]. Conversely, various liver diseases are often characterized by impaired intestinal barrier integrity and microbiota disruptions [9]. The gastrointestinal tract and liver are primary target organs for both the toxic effects and metabolism of DON. DON exposure disrupts gut microbiota balance and compromises intestinal barrier integrity, thereby increasing susceptibility to pathogen infection. Furthermore, DON can induce metabolic disorders and hepatic damage by triggering oxidative stress and inflammatory responses [17,18]. However, the specific role of gut microbiota in DON-induced hepatointestinal injury and the underlying mechanisms remain poorly understood. A thorough investigation into the complex interplay between the gut and liver in DON-induced pathology will facilitate the development of novel targeted therapeutic strategies.

Given the higher detection rate of DON in grain and feed, it is crucial to identify a novel feed additive that can effectively degrade DON and enhance immune function, thereby mitigating the adverse effects of DON on animals and reducing toxin residues. Due to the limitations of traditional physical and chemical methods, nutritional interventions, especially those utilizing plant extracts, have become a significant focus in recent detoxification research [19,20]. Chlorogenic acid (CGA) is a polyphenolic compound predominantly found in plants such as honeysuckle, eucommia, and coffee. Studies have proved that CGA possesses antioxidant, anti-inflammatory, and immuno-enhancing properties. These characteristics enable CGA to alleviate DON-induced intestinal damage by scavenging excess reactive oxygen species [21,22,23]. VX765 is a selective inhibitor that exhibits potent nutritional regulatory effects, as well as antioxidant and anti-inflammatory properties [24,25]. It holds substantial therapeutic potential for a broad spectrum of inflammation-related diseases. Multiple studies have reported that oxidative stress and inflammatory responses are pivotal mechanisms underlying DON-induced toxicity [26,27]. However, the therapeutic effects and potential mechanisms of CGA and VX765 on DON-induced toxicity remain unclear. Therefore, this study evaluated the therapeutic effects of CGA and VX765 on enterohepatic injury induced by DON using mice as animal models, thereby exploring their potential mechanisms from the perspective of the microbiota-gut-liver axis. This research will provide a theoretical foundation for the development of CGA and VX765 as nutritional health products, adjuvant drugs, and feed additives.

## 2. Results

### 2.1. CGA and VX765 Alleviate DON-Induced Intestinal Barrier Breakdown

Histological analysis (Figure 1a) revealed that the ileal and cecal epithelial cells in the control group had a well-organized structure and clear intestinal architecture. In contrast, the DON group exhibited disordered villus structures, along with atrophy and deformation of the intestinal glands. However, CGA and VX765 reduced these pathological changes. Compared with the control group, the DON group had significantly lower mRNA levels of *mucin2* (*MUC2*) (*p* < 0.001), *zonula occludens protein 1* (*ZO-1*) (*p* < 0.01), *occludin* (*p* < 0.001), and *claudin* (*p* < 0.001) in both the ileum and cecum (Figure 1b,c). CGA (*p* < 0.05) and VX765 (*p* < 0.05) partially reversed these reductions, except for ileal occludin expression, which did not change significantly (*p* > 0.05).

### 2.2. CGA and VX765 Intervene in the Intestinal Flora Disturbance Induced by DON

2bRAD-M sequencing was used to analyze the cecal microbiota in mice, and clean data were deposited in NCBI’s SRA library (PRJNA 1243859). Compared with the control group, the Chao1, Shannon, and Simpson indices in the DON group were significantly lower, indicating reduced microbial richness and diversity. These reductions were mitigated by CGA and VX765 treatment (Figure 2a). PCoA and PCA showed distinct clustering among the four groups, with partial overlap between CGA and VX765 groups (Figure 2b). Significant differences in cecal microbial composition were observed at the phylum (Figure 2c) and genus (Figure 3a) levels. *Firmicutes*, *Bacteroidetes*, *Campylobacterota*, and *Deferribacterota* were the dominant bacterial taxa, accounting for over 90% of the total microbiota. Notably, their abundances were significantly lower (*p* < 0.01) in the DON group than in the other three groups (Figure 2d). Additionally, the ratios of *Firmicutes* to *Bacteroidetes* (F/B) were significantly reduced in both the DON and VX765 groups compared to the control and CGA groups (*p* < 0.05), a shift associated with altered lipid metabolism (Figure 2e). *Deferribacterota* abundance was significantly higher (*p* < 0.05) in the DON group than in the other groups, and *Pseudomonadota* was elevated (*p* < 0.05) in both the DON and VX groups compared to the control group (Figure 2f,g).

Petal diagram showed that the DON group had the lowest total number of identified flora compared to the other three groups (Figure 3b). Venn diagram analysis of differential genera illustrated both distinct and overlapping bacterial features among the four groups (Figure 3c). Using ANOVA, we quantified the number of microorganisms showing significant differences (*p* < 0.05) across taxonomic levels in different treatment groups. Results indicated that although both CGA and VX765 helped reshape the DON-disrupted microflora, they exhibited both overlapping and unique effects (Figure 3d). LEfSe analysis (LDA score > 4 and *p* < 0.05) identified significantly different microbiota at the phylum and genus levels. As shown in Figure 3e, at the genus level, *Mucispirillum* and *Muribaculum* were significantly more abundant in the DON group, while *UBA3282*, *Pelethomonas*, *Eubacterium_J*, *Ventrimonas*, *Dysosmobacter*, and *Acutalibacter* were significantly reduced compared to the control. Except for *Eubacterium_J*, both CGA and VX765 reversed these changes. Additionally, both CGA and VX765 effectively counteracted DON-induced downregulation of *Marseille_P3106* and upregulation of *Fimiplasma* (Figure 3f). Random forest (Appendix A) and indicator analysis (Appendix A) identified key differential species across the four groups. Results showed that both DON and VX765 reversed DON-induced increases in *Mucispirillum* and *Paraprevotella*, as well as decreases in *Pelethomonas*, *Ventrimonas*, *Dysosmobacter*, and *Actutalibacter*.

In this study, KEGG functional pathways were enriched using PICRUSt2 predictions. Analysis showed that CGA (Figure 4a) and VX765 (Figure 4b) activated distinct regulatory pathways in response to DON-induced microbial metabolic disruption. The common pathways affected by both compounds included the phosphatidylinositol signaling system, meiosis-yeast, renal cell carcinoma, nonribosomal peptide structures, and retrograde endocannabinoid signaling (Figure 4c). Notably, DON significantly upregulated two lipid metabolism pathways: the phosphatidylinositol signaling system and retrograde endocannabinoid signaling (*p* < 0.05). Both CGA and VX765 effectively reversed this upregulation (*p* < 0.05) (Figure 4d,e). The Kruskal–Wallis test was also used to analyze protein Clusters of Orthologous Groups (COG). Results showed that enzymes linked to lipid metabolism, such as pyrophosphokinase, diphosphohydrolase, and methyltransferase, were significantly elevated in the DON group (*p* < 0.05). Both treatments reversed these changes, with CGA showing a stronger effect (Figure 4f).

### 2.3. CGA and VX765 Intervene in the Cecal Fungal Flora Induced by DON

This study evaluated the effects of different treatments on cecal fungal flora using 2bRAD-M sequencing. α-diversity analysis showed that fungal diversity in the DON group was significantly higher than that in the control group (Figure 5a). In contrast, the CGA group showed a significant increase in diversity compared to both the DON and control groups, while the VX group showed no significant changes. At the phylum level, *Basidiomycota* was more abundant in the DON and CGA groups than in the control and VX groups (Figure 5b). At the genus level, *Puccinia* was significantly more abundant in the DON and CGA groups compared to the control and VX groups. Further analysis of Puccinia species (Figure 5c), including *Striiformis* and *Triticina*, revealed that both were significantly more abundant in the DON and CGA groups, with *Triticina* completely absent in the VX group.

### 2.4. Correlation Analysis of Liver Lipid Metabolism and Cecal Microbiota

A total of 6 classes and 10 sub-classes, representing 2339 lipid metabolites, were identified by UPLC-MS/MS in this study (Appendix A). PCA scores (Figure 6a) showed consistent overall distribution among the four sample groups. PLS-DA scores (Figure 6b) clearly differentiated metabolite profiles across all groups. Using the criteria *p* < 0.05 and VIP > 1, differential metabolites were selected from the metabolite matrix (Appendix A). Log2(Fold Change) < 0 indicated down-regulated metabolites, while Log2(Fold Change) > 0 indicated upregulated metabolites. The number of differentially detected metabolites in each comparison group is shown in Figure 6c. The top 50 metabolites with significant differences were sorted by *p*-value and subjected to hierarchical clustering (Figure 6d). Among these, phosphatidylserine (PS) and ceramide (Cer) accounted for 24% and 18%, respectively, making them the two largest categories. The greatest differences were observed between the control and DON groups, while the CGA group was most similar to the control. KEGG enrichment analysis showed that multiple pathways were significantly linked to fat absorption, metabolism, and regulation (Figure 6e).

Notably, retrograde endocannabinoid signaling functions not only as a key pathway in hepatic lipid metabolism but also influences DON-induced cecal microbiota metabolism. These findings suggest a potential link between gut microbes and liver lipid metabolism. To explore this further, we integrated microbiome and lipid metabolomics data using mixOmics tools, visualized feature screening results, and conducted correlation analysis. At the phylum level (Figure 7a), *Pseudomonadota*, *Spirochaetota*, and *Bacillota* were strongly linked to lipid metabolism in both the control and DON groups. Comparisons showed that DON vs. CGA was correlated with *Deferribacterota*, *Spirochaetota*, *Verrucomicrobiota*, and *Basidiomycota*, while DON vs. VX correlated with *Spirochaetota*, *Pseudomonadota*, *Cyanobacteriota*, and *Bacillota* in lipid metabolism. Correlation analysis with Figure 2 indicated that *Deferribacterota* in the CGA group and *Pseudomonadota* in the VX group play key roles in regulating DON-induced lipid metabolic abnormalities. At the genus level (Figure 7b), *Eubacterium* and *Pelethomonas* were closely associated with lipid metabolism in both the control and DON groups. Furthermore, *Mucispirillum*, *Eubacterium*, and *Enterobacter* showed strong relationships with lipid metabolism in the DON and CGA groups. Notably, *Eubacterium* was significantly linked to lipid metabolism specifically in the DON and VX groups. Correlation analysis with Figure 3 further showed that *Mucispirillum* and *Eubacterium* in the CGA group, as well as *Eubacterium* in the VX group, are key regulators of DON-induced lipid metabolic disturbances.

### 2.5. CGA and VX765 Intervene in Gene Expression Related to Lipid Metabolism in the Mice Liver

As shown in Figure 8a, the DON group had significantly lower body weight than the control group (*p* < 0.05). Both the CGA and VX765 groups showed a higher body weight than the DON group, although only the former difference was statistically significant (*p* < 0.05). Compared to controls, the DON group exhibited significantly reduced (*p* < 0.05) activities of catalase (CAT) and glutathione peroxidase 4 (GPX4) (Figure 8b,c). While CGA supplementation effectively restored enzyme levels (*p* < 0.05), VX765 treatment had no significant effect (*p* > 0.05). These findings suggest that CGA alleviates DON-induced hepatic oxidative damage, while VX765 may be ineffective. HE staining showed that the liver tissue in the control group was well-organized. In contrast, the DON group showed granular degeneration, cellular swelling, nuclear membrane disruption, and even nuclear loss, all of which were alleviated by CGA and VX765 (Figure 8d). The mRNA (Figure 8e) and protein (Figure 8f,g) levels of Inositol Monophosphatase (IMPA), CDP-diacylglycerol synthase 2 (CDS2), Diacylglycerol Kinase Alpha (DGKA), NADH:Ubiquinone Oxidoreductase Core Subunit S8 (NDUFS8), and Mitogen-Activated Protein Kinase-1 (MAPK1) were significantly higher (*p* < 0.05) in the DON group than in the control group, and these increases were reversed by CGA and VX765 (*p* < 0.05). Additionally, CGA and VX765 reduced the DON-induced increase in IMPA and NDUFS8 fluorescence intensity (Figure 8h).

## 3. Discussion

Most monogastric animals are sensitive to chronic or subchronic exposure to DON 4, and DON can compromise intestinal function by increasing barrier permeability, inducing barrier damage, and altering microbial composition [28,29,30]. In our study, continuous gavage of 1 mg/kg DON in mice resulted in histological alterations in the ileum and cecum, as well as reduced mRNA expression levels of *MUC2*, *ZO-1*, *occludin* and *claudin*, indicating impaired barrier integrity and increased susceptibility to pathogenic infections.

In recent years, the rapid advancement of high-throughput sequencing technology has provided robust technical support for in-depth investigations into the intestinal microbiome. 2bRAD-M is a microbial diversity analysis method by employing type IIB restriction endonucleases to perform qualitative and semi-quantitative analyses of unique tags generated from microbial genomes after enzymatic digestion [31]. Compared to 16S/18S/ITS amplicon sequencing, 2bRAD-M exhibits notable advantages in detection range, resolution, sample throughput, accuracy, and sensitivity [32]. In this study, 2bRAD-M sequencing was used to analyze the cecal bacterial and fungal flora. The results showed that DON exposure reduced microbial richness and diversity of mice, disrupted intestinal microbiota balance, and significantly altered several bacterial genera, which may adversely affect host nutrient sensing and energy metabolism [33]. The addition of CGA and VX765 reversed the DON-induced reduction in bacterial diversity. Furthermore, DON exposure significantly down-regulated the F/B ratio, which is linked to lipid metabolism [34,35]. CGA effectively reversed this effect, whereas VX765 did not exhibit such capability. Random forest and indicator analysis revealed that both CGA and VX765 mitigated the DON-induced increases in *Mucispirillum* and *Paraprevotella*. The aberrant elevation of these bacteria may disrupt intestinal flora homeostasis, compromise intestinal barrier integrity, and consequently lead to lipid metabolism disturbances [36]. Additionally, CGA and VX765 reversed the DON-induced decrease in *Pelethomonas, Ventrimonas, Dysosmobacter, and Acutalibacter*. The stabilization of these bacteria contributes to maintaining the equilibrium of the gut microbial flora, enhancing intestinal barrier function, and regulating metabolic disturbances [37,38,39]. In addition, DON exposure significantly down-regulated F/B ratios, which could potentially impair host feeding behavior and energy metabolism [34]. Notably, CGA effectively reversed this effect, while VX765 failed to demonstrate such capability. In short, the addition of CGA and VX765 contributed to restoring the microbiota imbalance induced by DON and reshaping the structure of the intestinal microbiome. Despite significant differences in how CGA and VX765 modulate intestinal flora, both compounds can mitigate DON-induced intestinal barrier damage by targeting shared differential bacteria. PICRUSt2 is commonly employed to evaluate functional differences among various samples and groups based on the composition of known microbial gene functions [40]. In summary, CGA and VX765 help restore DON-induced microbiota dysbiosis and reshape the intestinal microbial structure, as shown by increased beneficial bacteria and reduced opportunistic pathogens. This microbial shift is important because a balanced gut microbiota supports metabolic activity and maintains intestinal barrier integrity. Although CGA and VX765 act through different mechanisms, both compounds improve intestinal barrier dysfunction by targeting shared bacterial taxa. These findings suggest that DON-induced metabolic disturbances may stem from microbiota dysbiosis, with inflammation likely resulting from barrier disruption and metabolic imbalance.

PICRUSt2 is widely utilized to evaluate functional variations among different samples and groups based on the composition of known microbial gene functions. KEGG enrichment in this study showed that DON exposure significantly activated the phosphatidylinositol and endocannabinoid signaling systems. These pathways have been recently identified as key regulators of lipid metabolism, and their dysregulation is closely associated with metabolic disorders such as obesity, insulin resistance, and non-alcoholic fatty liver disease (NAFLD) [41,42]. Lipid metabolism is essential for cellular energy and membrane stability. Our findings revealed that DON-induced microbiota dysbiosis is closely linked to disturbances in lipid metabolism, and both CGA and VX765 can reduce these effects by modulating the microbiota.

The liver is the primary target organ for the metabolic and toxic effects of DON, which can induce liver damage through oxidative stress and inflammatory responses [17,43]. In this study, the addition of CGA and VX765 was found to attenuate DON-induced liver tissue injury. The key distinction lies in the fact that CGA can alleviate DON-induced oxidative damage, whereas VX765 does not demonstrate this effect, likely attributable to the differing mechanisms of action between the two compounds. The liver serves as a pivotal organ in lipid metabolism, functioning as the primary site for lipid synthesis, transport, metabolism, and storage [44]. In this study, UPLC-MS/MS was employed to assess the lipid composition of the liver in each treatment group. The findings revealed that PS and Cer were the two major metabolites influenced by DON and modulated by CGA and VX765. Additionally, several pathways associated with fat absorption, metabolism, and regulation were affected by DON and regulated by CGA and VX765, indicating that CGA and VX765 can ameliorate DON-induced abnormalities in hepatic lipid metabolism. Notably, retrograde endocannabinoid signaling not only represented a critical pathway through which CGA and VX765 regulate hepatic lipid metabolism but also played a key role in microbial metabolism within the cecum. These results suggested a broad and intricate relationship between the liver and gut, wherein microbial metabolism acts as a central mediator of this interaction [45,46]. Combined microbiome and lipid metabolomics analysis revealed that *Deferribacterota* in the CGA group and *Pseudomonas* in the VX group, along with *Mucispirillum* and *Eubacterium* in the CGA group and *Eubacterium* in the VX group, were key contributors to modulating DON-induced abnormalities in lipid metabolism.

By integrating KEGG analysis results with gene association data from the GeneCards database, we identified a strong correlation between *DGKA, MAPK1*, and the roles of DON, CGA and VX765. Additionally, *IMPA*, *CDS2*, *DGKA*, and *NDUFS8* were found to be closely associated with these two genes, all of which play significant roles in hepatic lipid metabolism (Appendix A). Our findings proved that DON exposure alters hepatic lipid metabolism through the upregulation of mRNA and protein expression levels of IMPA, CDS2, DGKA, NDUFS8 and MAPK1 (Appendix A). IMPA is integral to the synthesis and metabolism of inositol, thereby influencing the normal physiological functions of cells [47]. CDS2 participates in the biosynthesis and homeostasis of intracellular phospholipids, consequently affecting cellular membrane structure and function [48]. DGKA catalyzes the phosphorylation of diacylglycerol into phosphatidic acid, a process critical for cell signaling and lipid metabolism [49,50]. NDUFS8 plays a pivotal role in regulating energy and lipid metabolism by contributing to mitochondrial respiratory chain function [51]. Aberrant expression of these genes, as observed in this study, may be associated with DON-induced liver injury and alterations in lipid metabolism. Nevertheless, both CGA and VX765 have demonstrated the ability to mitigate DON-induced hepatic injury and lipid metabolic disorders. In summary, CGA could reverse liver oxidative damage and weight loss induced by DON in mice. VX765 alleviated DON-induced liver damage and lipid metabolism disorders, but did not affect oxidative damage or weight loss. Both CGA and VX765 mitigated DON-induced hepatic lipid metabolism issues, though via different mechanisms. CGA, as a plant-derived antioxidant, may be more suitable than VX765 for controlling DON effects.

The “microbiota-gut-liver axis” represents a highly conserved physiological mechanism in mammals [52]. DON harms the gut barrier, liver, and gut microbiota, as shown in human and animal studies. The detrimental effects of DON on the intestinal barrier, liver function, and microbial composition have been extensively documented in both human and animal studies. Therefore, targeting the microbiota to protect this axis in humans is physiologically sound. CGA is a natural compound found in coffee, fruits, and vegetables, and is safe for humans. As a dietary supplementation strategy, CGA offers greater acceptability and practicality compared to pharmacological agents. However, human gut microbiota are more diverse than those in mice and are affected by genetics, diet, environment, and medications like antibiotics [53]. Mice are usually exposed to a fixed dose of DON in controlled labs, but humans face long-term, low-dose exposure along with other toxins. Translating mouse doses to humans also needs careful calculation and clinical testing. VX765’s limited effectiveness in mice suggests that targeted anti-inflammatory therapies focusing on a single pathway may be insufficient for addressing the multi-systemic effects of DON.

## 4. Materials and Methods

### 4.1. Reagents

CGA (C_16_H_18_O_9_, purity > 98%) was purchased from Macklin Company (Shanghai, China), VX765 (C_24_H_33_ClN_4_O_6_, purity > 98%) was obtained from AbMole Company (Houston, TX, USA), and DON (C_15_H_20_O_6_, purity > 98%) was acquired from Triplebond Company (Guelph, ON, Canada). After being dissolved in DMSO, the mother liquors of CGA, VX765, and DON were prepared and stored at −80 °C. Subsequently, they were diluted with normal saline to the desired concentration prior to experimentation.

### 4.2. Animal, Experimental Design and Sampling

48 SPF female KM mice (28 ± 2 g, *p* > 0.05) with 5-week-old were equally divided into four groups: Control (normal saline), DON (1 mg/kg.bw), CGA (100 g/kg.bw CGA + 1 mg/kg.bw DON), and VX groups (100 mg/kg.bw VX765 + 1 mg/kg.bw DON). Following a 7-day acclimation period, all mice received drug administration via fasting gavage at a dosage volume of 0.2 mL per day (Appendix A). The standard diet was provided by Pengyue Experimental Animal Breeding Co., LTD. (Jinan, China), and the detailed nutritional composition is shown in Appendix A. The experimental duration was 28 days in total, during which all mice were housed in a controlled environment with a constant temperature of 23 °C, humidity ranging from 40% to 60%, and a 12 h light/dark cycle. On day 29, all experimental mice were euthanized via cervical dislocation. Tissue samples from the liver, ileum, and cecum were rapidly excised; some were stored at −80 °C, while others were fixed in a 4% paraformaldehyde solution for subsequent analysis. Additionally, cecal contents were preserved at −80 °C for further processing.

### 4.3. Hematoxylin-Eosin (HE) Staining

Following a 24 h rinse with running water, the fixed tissue samples underwent gradient alcohol dehydration, xylene clarification, and paraffin infiltration to prepare histological sections. The sections were dewaxed with xylene and rehydrated using gradient ethanol, then stained with hematoxylin for 5–10 min, lightly rinsed with running water, differentiated with 1% hydrochloric acid in ethanol for a few seconds, and blued using a bluing agent. Subsequently, the sections were stained with eosin for 1–3 min, after which the staining reaction was terminated by rinsing with running water. Finally, the sections were dehydrated using gradient ethanol, cleared with xylene, and mounted with neutral gum. Histological evaluation was conducted under an optical microscope (Nikon, Tokyo, Japan).

### 4.4. qRT-PCR

Tissue samples were homogenized with Trizol (Invitrogen, Carlsbad, CA, USA), followed by chloroform phase separation, isopropanol precipitation, and ethanol washing to remove contaminants. Total RNA was then dissolved in DEPC-treated, RNase-free water. RNA concentration and purity were assessed using a spectrophotometer (DeNovix, Wilmington, DE, USA) based on the A260/A280. Subsequently, RNA was reverse-transcribed into cDNA following the protocol for Evo M-MLV RT Mix Kit (LongGen, Hangzhou, China). All primers were synthesized by Sangon Biotech (Shanghai, China), with sequences listed in Table 1. The resulting cDNA was used as a template for qRT-PCR with SYBR^®^ Green I premix on a fluorescence quantitative PCR system (Applied Biosystems, South San Francisco, CA, USA). The thermal cycling program included initial denaturation at 95 °C for 5 min, followed by 40 cycles of denaturation at 95 °C for 10 s and annealing/extension at 60 °C for 30 s. Amplification specificity was confirmed by melting curve analysis, and target gene expression was calculated using the 2^−ΔΔCt^ method, normalized to the reference gene *GAPDH*.

### 4.5. Western Blotting (WB)

The tissue samples were ground and lysed in lysis buffer containing protease inhibitors, and the supernatant was collected after centrifugation. Protein concentration was measured using the BCA method, and a standard curve was generated to determine sample concentrations. Equal amounts of protein (30 μg) were mixed with 5 × SDS loading buffer, denatured at 95 °C for 5 min, and separated by 10% SDS-PAGE. Proteins were then transferred to a 0.45 μm polyvinylidene fluoride (PVDF) membrane (Millipore, Millipore, Burlington, MA, USA) at 200 mA for 90 min. The membrane was blocked with 5% skim milk at room temperature for 2 h, then incubated with the primary antibody at 4 °C overnight. After washing with TBST, the membrane was incubated with HRP-labeled secondary antibody at room temperature for 2 h. Following another wash, the membrane was developed using ECL chemiluminescent reagent and imaged with the Fusion FX7 Vilber Lourmat system (Vilber Lourmat, Marne-la-Vallée, France). Protein bands were quantified using ImageJ software 1.54f (NIH, Bethesda, MD, USA). The expression of the target protein was normalized to β-actin. Detailed information regarding the antibodies is provided in Appendix A.

### 4.6. Immunofluorescence (IF)

Liver tissue samples were embedded in paraffin and sectioned (5 μm) onto poly-L-lysine-coated slides, which were subsequently baked at 60 °C for 30 min. Next, Antigen retrieval was performed using sodium citrate buffer (pH 6.0) via heat-mediated methods. Sections were rinsed with phosphate-buffered saline (PBS) and permeabilized with 0.1–0.5% Triton X-100 for 10 min to facilitate antibody penetration. Blocking was carried out with 5% BSA at room temperature for 1 h to reduce non-specific binding. The primary antibody was applied, and slides were incubated overnight at 4 °C. After PBS washing, the sections were incubated with HRP-conjugated secondary antibody at room temperature for 1 h in the dark. Nuclei were counterstained with DAPI (1 μg/mL) for 5–10 min in the dark, followed by PBS rinsing and mounting with an anti-fade medium. Fluorescent images were captured promptly using a confocal fluorescence microscope (Nikon, Tokyo, Japan) and analyzed for fluorescence intensity and localization with ImageJ software 1.54f (NIH, Bethesda, MD, USA).

### 4.7. UPLC-MS/MS

The lipid composition of tissue was analyzed qualitatively and quantitatively by UPLC-MS/MS. Liver samples were thawed at room temperature, weighed, and added to a 2:1 chloroform-methanol mixture containing 0.1 mM BHT. The mixtures were then thoroughly vortexed. After homogenization, ultrasonic-assisted extraction, and centrifugation, a 5 μL aliquot of the supernatant was injected into the UPLC system. The elution program was as follows: 0 min, 30% B; 3 min, 30% B; 5 min, 62% B; 15 min, 82% B; 16.5 min, 99% B; 18 min, 99% B; 18.1 min, 30% B; 22 min, 30% B. After elution, the lipid components were introduced into the mass spectrometer for ionization and detection, with multiple reaction monitoring (MRM) data being collected in real time. The lipid types in the sample were identified by matching their retention times and MRM ion pair information against reference standards or databases. A calibration curve was constructed using known concentrations of internal standards to accurately quantify the content of each lipid species in the sample. According to the parent ion and multistage mass spectrometry data of each sample, the original data matrix was systematically constructed. Principal component analysis (PCA) was applied to evaluate sample stability and distribution based on this matrix, and supervised partial least squares discriminant analysis (PLS-DA) was used to identify metabolic profile differences among groups.

### 4.8. 2bRAD-M

2bRAD-M technique was employed to assess gut microbiota structure and potential metabolic pathways. Genomic DNA was extracted from cecal microorganisms using the Micro DNA Kit (Tiangen, Beijing, China). DNA was digested with BcgI (NEB, Beijing, China) at 37 °C for 3 h and ligated to library-specific adapters. After PCR amplification and purification, sequencing was performed on the Illumina Nova PE150 platform (Oebiotech, Qingdao, China). A total of 404,199 microbial genomes (bacteria, fungi, and archaea) were obtained from the GTDB and Ensembl databases. Each genome was digested with 16 type 2b restriction enzymes to generate unique species-specific 2bRAD tags, ensuring no overlap within the same taxonomic group. These tags were utilized to build a comprehensive 2bRAD microbial genome database. After quality filtering, all 2bRAD tags were mapped to the database for taxonomic annotation and relative abundance analysis. α-diversity indices (Shannon, chao1, Simpson) were calculated using the R software package (version 4.3.3) [54]. β-diversity was assessed via Binary Jaccard distance with Adonis tests, using both PCA and principal coordinate analysis (PCoA). Furthermore, microbial gene function was predicted using PICRUSt2 (version 2.3.0b0).

### 4.9. Statistical Analysis

After pre-processing the data in Excel, one-way ANOVA was performed using GraphPad Prism 8.0 for Windows (GraphPad Software, San Diego, CA, USA). Duncan’s test was used to analyze group differences. Results are presented as mean ± SD, with significance denoted by *p* < 0.05 and extreme significance by *p* < 0.01. 

## 5. Conclusions

In conclusion, DON exposure caused cecal microbiota dysbiosis in mice, disrupted intestinal barrier function, and triggered liver damage and lipid metabolism disorders via the microbiota–gut–liver axis. As nutritional additives, CGA and VX765 alleviated DON-induced enterohepatic injury and lipid metabolic disturbances by modulating the gut microbiota composition and the microbiota–gut–liver axis. Notably, CGA demonstrated a stronger protective effect against DON-induced enterohepatic injury than VX765.

## Figures and Tables

**Figure 1 toxins-17-00467-f001:**
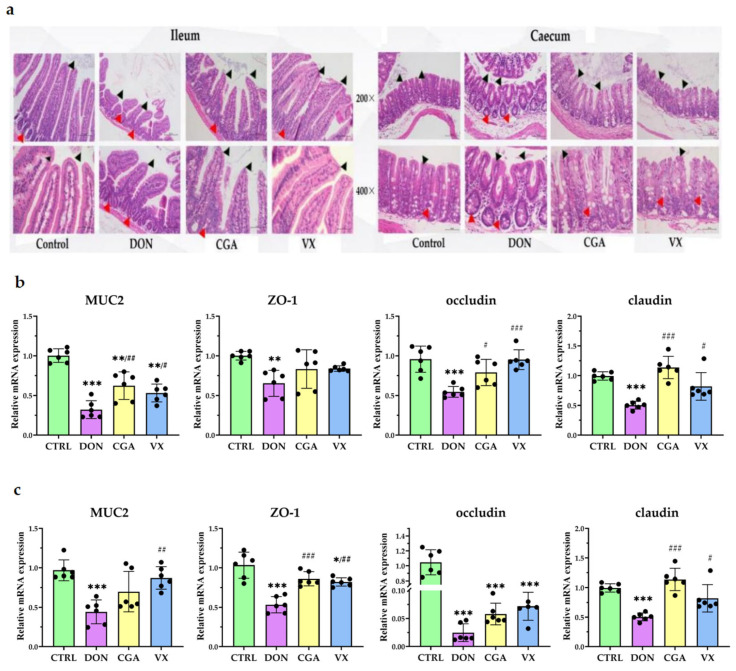
CGA and VX765 alleviated DON-induced intestinal barrier injury. (**a**) HE staining images of the ileum and cecum (*n* = 3). Black arrows indicate villus rupture; red arrows indicate intestinal gland damage. mRNA expression levels of *MUC2*, *ZO-1*, *occludin*, and *claudin* in the ileum (**b**) and caecum (**c**), mean ± SD, *n* = 6. *, **, *** indicate *p* < 0.05, 0.01, 0.001 vs. Control group; ^#^, ^##^, ^###^ indicate *p* < 0.05, 0.01, 0.001 vs. DON group.

**Figure 2 toxins-17-00467-f002:**
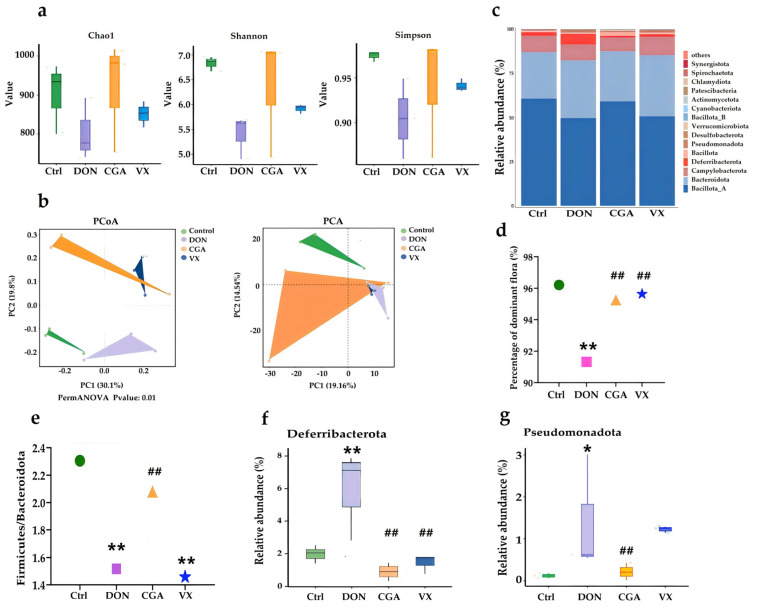
Effects of CGA and VX765 on DON-induced dysbiosis of the cecal microbiota. (**a**) Alpha diversity analysis, including chao1, shannon and simpson indices. (**b**) Beta diversity analysis, including PCoA and PCA. (**c**) Relative abundance of bacteria composition at the phylum level (top 15). (**d**) Percent of dominant phyla, including *Firmicutes*, *Bacteroidetes*, *Campylobacterota*, and *Deferribacterota*. (**e**) F/B ratio. Relative abundance of *Deferribacterota* (**f**) and *Pseudomonadota* (**g**) Green, purple, yellow, and blue represent control, DON, CGA, and VX groups, respectively. *, ** indicate *p* < 0.05, 0.01 vs. control; ^##^ indicate *p* < 0.01 vs. DON group. F/B ratio: the ratio of *Firmicutes* to *Bacteroidetes*.

**Figure 3 toxins-17-00467-f003:**
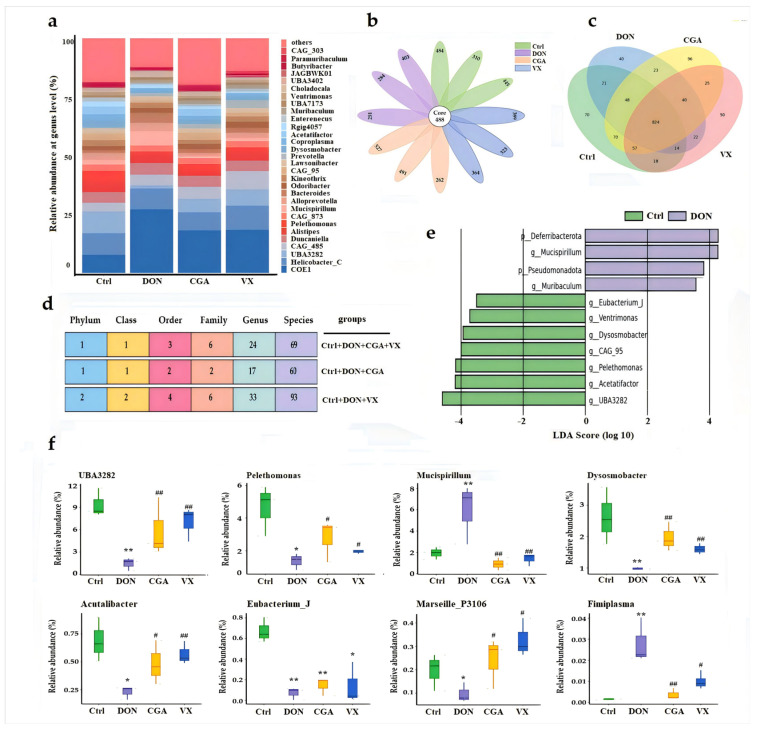
Analysis of bacterial composition at the genus level based on 2bRAD-M sequencing (*n* = 3). (**a**) Relative abundance of the top 30 genera. (**b**) Petal diagram: inner numbers indicate species common to all groups; outer numbers show unique species per group. (**c**) Venn diagram of differential genera. (**d**) ANOVA analysis of differential species across taxonomic levels. (**e**) LEfSe analysis identifying significant phylum- and genus-level differences (LDA score > 4 and *p* < 0.05). (**f**) Several bacterial species with significant changes. *, ** indicate *p* < 0.05, 0.01 vs. control; ^#^, ^##^ indicate *p* < 0.05, 0.01 vs. DON group. LEfSe: Linear discriminant analysis Effect Size.

**Figure 4 toxins-17-00467-f004:**
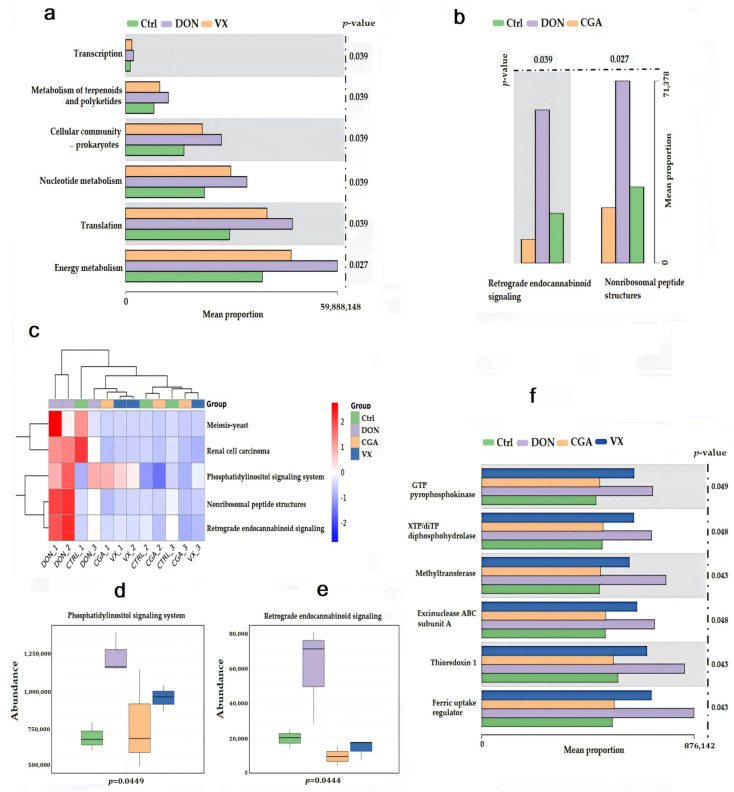
Effects of DON, CGA, and VX765 on cecal flora function based on PICRUSt2 predictions. (**a**) KEGG-enriched pathways in the Control, DON, and VX groups. (**b**) KEGG-enriched pathways in the Control, DON, and CGA groups. (**c**) KEGG-enriched pathways in the Control, DON, CGA, and VX groups. Abundance of the phosphatidylinositol signaling system (**d**) and retrograde endocannabinoid signaling (**e**) across the four groups. (**f**) Bar chart showing differences in COG results, illustrating the average pathway abundance per group.

**Figure 5 toxins-17-00467-f005:**
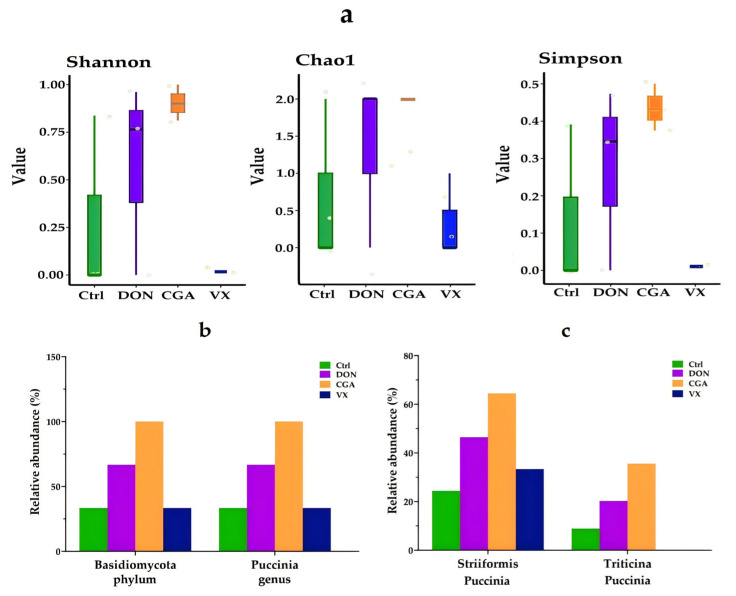
Effects of DON, CGA and VX765 on DON-induced dysbiosis of the cecal fungal flora. (**a**) Alpha diversity analysis, including Shannon, Chao1 and Simpson index. (**b**) Relative abundance of fungal composition at the phylum level and genus level. (**c**) Relative abundance of Striiformis and triticina.

**Figure 6 toxins-17-00467-f006:**
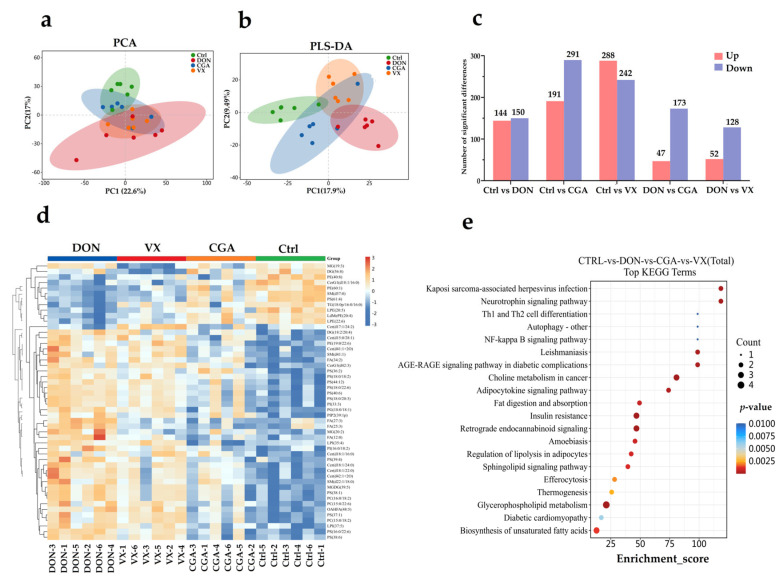
Effects of DON, CGA, and VX765 on the hepatic lipidomic profile of mice based on UPLC-MS/MS (*n* = 6). (**a**) PCA score plot. (**b**) PLS-DA score plot. (**c**) Number of differential metabolites in each comparison group. (**d**) Heatmap of the top 50 differentially expressed metabolites. The *x*-axis shows sample names; the *y*-axis lists the metabolites. Color intensity from blue to red reflects low to high expression. (**e**) KEGG enrichment analysis of differential metabolites (top 20). The horizontal axis shows the enrichment score; the vertical axis lists the top 20 pathways. Larger, redder bubbles indicate greater significance and more enriched metabolites.

**Figure 7 toxins-17-00467-f007:**
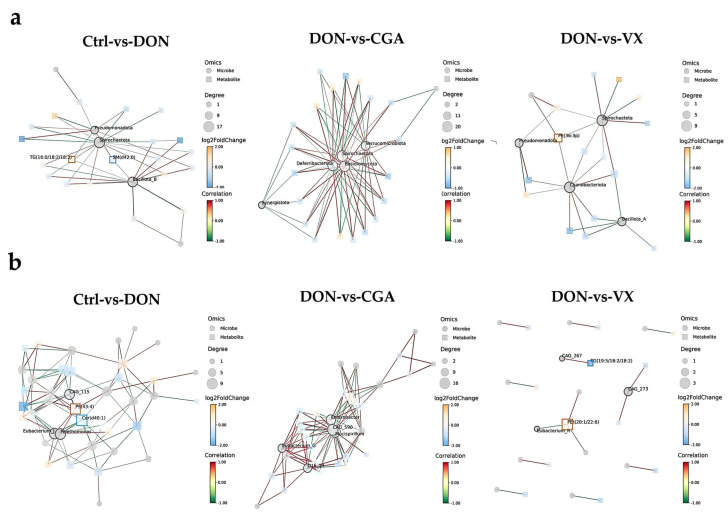
Network diagram of correlation analysis between differential microorganisms and differential lipid metabolites. (**a**) At phylum level. (**b**) At genus level.

**Figure 8 toxins-17-00467-f008:**
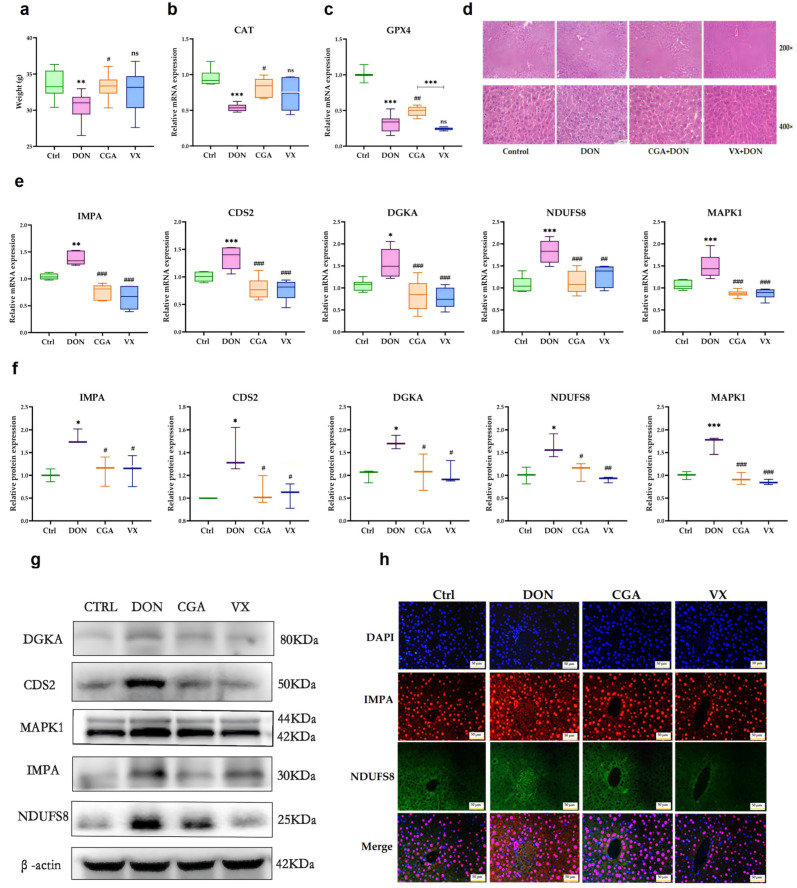
Effects of DON, CGA, and VX765 on lipid metabolism-related gene expression in mouse liver. (**a**) Body weight (mean ± SD, *n* = 12). Relative mRNA levels of (**b**) *CAT* and (**c**) *GPX4* (mean ± SD, *n* = 6). (**d**) HE-stained liver tissue images. (**e**) Relative mRNA expression of *IMPA*, *CDS2*, *DGKA*, *NDUFS8*, and *MAPK1* (mean ± SD, *n* = 6). Relative protein levels (**f**) and grayscale images (**g**) of IMPA, CDS2, DGKA, NDUFS8, and MAPK1 (mean ± SD, *n* = 3). (**h**) Representative immunofluorescence images of IMPA and NDUFS8. ns, *, **, *** denote *p* < 0.05; *p* < 0.01; *p* < 0.001 vs. control group; ^#^, ^##^, ^###^ denote *p* < 0.05; *p* < 0.01; *p* < 0.001 vs. DON group.

**Table 1 toxins-17-00467-t001:** The primer sequence for qRT-PCR analysis used in this study.

Gene		Sequence	Length (bp)	Gene ID	GenBank No.
*Mapk1*	F ^1^	CAGGTGTTCGACGTAGGGC	139	26413	NM_011949.3
	R ^2^	TCTGGTGCTCAAAAGGACTGA			
*GPX4*	F	GCAGGAGCCAGGAAGTAATCAAG	112	625249	NM_001037741.4
	R	ACAGTGGGTGGGCATCGTC			
*CTA*	F	CGTCCGTCCCTGCTGTCTC	129	12359	NM_009804.2
	R	GCTCCTTCCACTGCTTCATCTG			
*NDUFS8*	F	TGCAAACTCTGTGAGGCCAT	94	225887	NM_144870.5
	R	TGTCATAGCGTGTCGTTCGG			
*IMPA*	F	GCTGCTGTTAATATGTGCCTTGTG	110	55980	NM_018864.7
	R	CCTGCCTCGGTGACAATGATG			
*DGKA*	F	GCCACATCTGAGTCCATCTTCT	120	13139	NM_016811.4
	R	GTTCAATACCGCAATGCCTTCT			
*CDS2*	F	ACCAACCTCCGTAGATGACAC	87	110911	NM_138651.7
	R	CCTCTCACCCACCAGTTCTTC			
*MUC2*	F	ACCTCACAAGCAGTATCAGGC	136	17831	NM_023566.4
	R	GTCATAGCCAGGGGCAAACT			
*Claudin*	F	TAAGGCACGGGTAGCACTCAC	88	12741	NM_009386.3
	R	GATGTTGGCGAACCAGCAGAG			
*ZO-1*	F	TCCCACAAGGAGCCATTCCTG	115	21872	NM_009386.3
	R	GGGCTCAGCAGAGTTTCACCT			
*Occludin*	F	AGGAGGACTGGGTCAGGGAATA	123	18260	NM_008756.2
	R	TGACGTCGTCTAGTTCTGCCTG			
*GAPDH*	F	TGTGTCCGTCGTGGATCTGA	150	14433	NM_001289726.2
	R	TTGCTGTTGAAGTCGCAGGAG			

^1,2^ F means forward primer (5′ to 3′ direction), and R means reverse primer (5′ to 3′ direction).

## Data Availability

The datasets supporting the conclusions of this article are included within the article (and its additional files). 2bRAD-M sequencing cleaned data has been uploaded to NCBI’s SRA library, and BioProject number PRJNA 1243859.

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
