# Peer review of "Chlorogenic Acid and VX765 Alleviate Deoxynivalenol-Induced Enterohepatic Injury and Lipid Metabolism Disorders by Improving Intestinal Microecology"

_toxins, 2025, doi:10.3390/toxins17090467_

Round 1
Reviewer 1 Report
Comments and Suggestions for Authors
The manuscript has improvements to be included in the M&M section, Results, Figures, and Discussion. Citations are not correct in the entire text and are all hidden, which makes it difficult to identify if the latest studies have been included accordingly.
M&M: Please provide all types of details for the primers used (ot as Supplementary tables but as tables included in the text), for the concentration of the reagents used (sections 2.3, 2.4, and 2.6). Most of the assays are not described and the methodology is lacking and poor for any researcher who wants to understand the procedure. Some of the assays are not even describing their interest.
Results and Figures: The number of figures is very high and needs to be reduced or simplified. Among that, the description of the results is tedious and makes it difficult to follow at some point and the quality of the figures is very poor. The information provided is so high that it makes it difficult to maintain focus on the study. So it needs to be split or simplified. Please, make sure that the inclusion of the Figures with the legend is clear, and write in italics the significance. Legend in figures is too long and confusing. This is because the amount of information is very high but not well-organized.
Discussion: According to the results obtained, the discussion is very short and does not compare or contrast the results with previous publications properly. This needs to be improved, all of it.
In summary, the study seems to be good, but the information is too long and not well-organized so it need to be reduced and showed clearly to provide the results and the information accordingly otherwise the focus of the research is missed.
Author Response
Dear Reviewer:
Great thanks for your and reviewers’ helpful and professional suggestions and the comments on our manuscript “Chlorogenic acid and VX765 alleviate deoxynivalenol-induced enterohepatic injury and lipid metabolism disorders by improving intestinal microecology” (toxins-3760195). Those comments are all valuable and very helpful for revising and improving our paper, as well as the important guiding significance to our researches. We have studied comments carefully and made corrections, and the modified part were highlighted in red font in the manuscript. In response to the questions raised by the editors and reviewers, we answered them one by one in the comment section.
Reviewer #1:
- The manuscript has improvements to be included in the M&M section, Results, Figures, and Discussion. Citations are not correct in the entire text and are all hidden, which makes it difficult to identify if the latest studies have been included accordingly.
--- We sincerely appreciate your professional review. In the revised manuscript, the Methods and Materials section, Results, Figures, and Discussion have all been updated accordingly. The references have also been revised to meet the journal’s formatting requirements. All revised sections are highlighted in red.
- M&M: Please provide all types of details for the primers used ( as Supplementary tables but as tables included in the text), for the concentration of the reagents used (sections 2.3, 2.4, and 2.6). Most of the assays are not described and the methodology is lacking and poor for any researcher who wants to understand the procedure. Some of the assays are not even describing their interest.
---- We sincerely appreciate your highly professional review. In accordance with your valuable suggestions, we have revised the Methods and Materials section, particularly Sections 2.3, 2.4, and 2.6. All revised content has been marked in red font.
- Results and Figures: The number of figures is very high and needs to be reduced or simplified. Among that, the description of the results is tedious and makes it difficult to follow at some point and the quality of the figures is very poor. The information provided is so high that it makes it difficult to maintain focus on the study. So it needs to be split or simplified. Please, make sure that the inclusion of the Figures with the legend is clear, and write in italics the significance. Legend in figures is too long and confusing. This is because the amount of information is very high but not well-organized.
---- We sincerely appreciate your highly professional review. In response to your comments, we have revised the Results and Charts sections. Specifically, we have split and simplified certain figures and streamlined the corresponding figure legends for improved clarity.
- Discussion: According to the results obtained, the discussion is very short and does not compare or contrast the results with previous publications properly. This needs to be improved, all of it.
---- Thank you very much for your very valuable suggestions. In the revised manuscript, we have updated the Results section and incorporated additional content. All additions are highlighted in red font.

Reviewer 2 Report
Comments and Suggestions for Authors
The manuscript fives a very detailed description of the methodology and results of the experiment to assess the effects of using CGA and VX765 to mitigate the effects of DON in mice. Yet it reads pleasantly, which is a compliment to the authors. I only have some minor, mainly textual comments.
Just as for chlorogenic acid, I would start with the full name (Belnacasan according to Wikipedia), and then move on with the shortname (VX765).
Check the consistency of terms:
chlorogenic acid (= without initial capital)
are “axis”(54), enterohepatic axis (62), microbe-liver axis (91) the same? If so, better use one term
VX765, and not VX or ‘VX 765’
92: ‘provides a foundation ..’. Don’t be too modest. In the discussion you can write more about the (dis)similarities between mice and men.
185: a reference is needed for calculating the 3 diversity indices. Besides, either standard R software was applied (and then supply the version number), or indeed a package (and then supply the name and version of the package).
309: where does ‘which’ refer to?
439: I would formulate the sentence in terms of: both the CGA and VX765 group showed a higher body weight than the DON-group, although only the former difference was statistically significant.
444: the result for the VX765 group is inconclusive, so I would write ‘may be’ instead of ‘is’.
472: add ‘of mice’
I googled ‘DON and chlorogenic acid’ and the 1st hit (‘https://doi.org/10.1016/j.biopha.2023.116003’) was not included in the reference list. So, please make a final check.
Comments on the Quality of English Language
The manuscript fives a very detailed description of the methodology and results of the experiment to assess the effects of using CGA and VX765 to mitigate the effects of DON in mice. Yet it reads pleasantly, which is a compliment to the authors. I only have some minor, mainly textual comments, that I supplied to the authors.
Author Response
Great thanks for your and reviewers’ helpful and professional suggestions and the comments on our manuscript “Chlorogenic acid and VX765 alleviate deoxynivalenol-induced enterohepatic injury and lipid metabolism disorders by improving intestinal microecology” (toxins-3760195). Those comments are all valuable and very helpful for revising and improving our paper, as well as the important guiding significance to our researches. We have studied comments carefully and made corrections, and the modified part were highlighted in red font in the manuscript. In response to the questions raised by the editors and reviewers, we answered them one by one in the comment section.
Reviewer #2:
- Just as for chlorogenic acid, I would start with the full name (Belnacasan according to Wikipedia), and then move on with the shortname (VX765).Check the consistency of terms: chlorogenic acid (= without initial capital) are “axis”(54), enterohepatic axis (62), microbe-liver axis (91) the same? If so, better use one term VX765, and not VX or ‘VX 765’
---- We sincerely appreciate your professional advice. In the revised manuscript, several key technical terms have been standardized in accordance with academic conventions.
92: ‘provides a foundation ..’. Don’t be too modest. In the discussion you can write more about the (dis)similarities between mice and men.
---- We sincerely appreciate your encouragement and guidance. In accordance with your suggestion, we have included additional content regarding the differences between humans and mice in the revised manuscript. All revised content has been marked in red font.
185: a reference is needed for calculating the 3 diversity indices. Besides, either standard R software was applied (and then supply the version number), or indeed a package (and then supply the name and version of the package).
---- Thank you very much for your professional advice. In the revised manuscript, we have added the relevant references and updated the software version for the α- diversity indices.
309: where does ‘which’ refer to?
---- We sincerely apologize for the previous lack of clarity in the language. In the revised draft, the language has been carefully revised for improved clarity and precision.
439: I would formulate the sentence in terms of: both the CGA and VX765 group showed a higher body weight than the DON-group, although only the former difference was statistically significant.
---- We sincerely appreciate your valuable suggestions. The language in this section has been revised in the revised draft and highlighted in red.
444: the result for the VX765 group is inconclusive, so I would write ‘may be’ instead of ‘is’.
---- Thank you very much for your suggestion. The language in this part has been revised in the new draft.
472: add ‘of mice’
---- Thank you very much for your suggestion. The language in this part has been revised in the new draft.
I googled ‘DON and chlorogenic acid’ and the 1st hit (‘https://doi.org/10.1016/j.biopha.2023.116003’) was not included in the reference list. So, please make a final check. https://doi.org/10.2174/1874467213999200819145942.
---- Thank you very much for your careful review. The references section in the new manuscript has been revised and checked.
